# Neck Circumference as a Predictor of Metabolic Syndrome in Koreans: A Cross-Sectional Study

**DOI:** 10.3390/nu13093029

**Published:** 2021-08-30

**Authors:** Kyoung-Yun Kim, Ha-Rin Moon, Jung-Mi Yun

**Affiliations:** 1Department of Clinical Nutrition, Sun-Han Hospital, Gwangju 61917, Korea; kimkyjs0906@gmail.com; 2Department of Food and Nutrition, Chonnam National University, Gwangju 61186, Korea; cjsghk1970@naver.com

**Keywords:** neck circumference, MetS, ROC curve, waist circumference, BMI, KNHANES

## Abstract

Metabolic syndrome (MetS) is a complex metabolic disorder and a high-risk condition for type 2 diabetes and cardiovascular disease. Rapid screening of at-risk individuals using accurate and time-saving tools is effective in disease management. Using the Korea National Health and Nutrition Examination Survey (KNHANES) data, we collected data from 2234 participants suitable for the study design, of which 974 (43.6%) were men and 1260 (56.4%) were women. We used receiver operating characteristic (ROC) curve analysis to estimate the optimal sex-specific neck circumference (NC) cut-off point to predict the MetS risk. To analyze the risk of MetS according to the estimated NC, logistic regression analysis was performed to identify the confounding factors. The result of the ROC analysis showed that the optimal neck cut-off points for predicting the risk of MetS were 38.25 cm (AUC: 0.759, 95% CI: 0.729–0.790) in men and 33.65 cm (AUC: 0.811, 95% CI: 0.782–0.840) in women. In the upper NC cut-off point compared to the lower NC cut-off point, NC was associated with an increased MetS risk by 2.014-fold (*p* = 0.010) in men and 3.650-fold (*p* < 0.001) in women, after adjustments. The current study supports NC as an effective anthropometric indicator for predicting the risk of MetS. It is suggested that more studies should be conducted to analyze the disease prediction effect of the combined application of anthropometric indicators currently in use and NC.

## 1. Introduction

Metabolic syndrome (MetS) is defined as a complex metabolic abnormality characterized by hyperglycemia, hypertriglyceridemia, dyslipidemia, abdominal obesity, and elevated blood pressure (BP), indicating a high-risk condition for type 2 diabetes and cardiovascular disease [1]. The diagnostic criteria have been proposed and modified several times by public health agencies. Currently, the most widely used definitions of MetS are those of the National Cholesterol Education Program Adult Treatment Panel III (NCEP-ATP III) and the International Diabetes Federation (IDF), with particular focus on waist circumference as a proxy measure for abdominal obesity [2]. The IDF recommends a waist circumference threshold of 94 cm for men and 80 cm for women for Europeans, and the American Heart Association/National Heart, Lung, and Blood Institute (AHA/NHLBI) recommend cut-off points of 102 cm and 88 cm, respectively. The IDF recognized the difficulty in identifying unified criteria for the MetS applicable to all ethnic groups and proposed a new criterion that included ethnic specificity [3]. This is to diagnose decisive MetS by considering ethnicity-specificity and to define the risk of type 2 diabetes and cardiovascular disease. Compared to Europeans, Asians have a much lower waist circumference (WC) threshold for abdominal obesity, considering the risk of type 2 diabetes [3,4]. The Asia-Pacific criterion for abdominal obesity, which has been used since 2006, is ≥90 cm in men and ≥80 cm in women [5].

According to this criterion, the mean waist circumference of Korean women is 78.6 cm, which is close to the criterion for diagnosing abdominal obesity. As such, re-evaluated diagnostic criteria of ≥90 cm in men and ≥85 cm in women are being used [6]. In future, this diagnostic definition could be modified according to the ability of disease indicators to accurately predict metabolic abnormalities associated with type 2 diabetes and cardiovascular disease [7].

As Alberti et al. suggested, waist circumference thresholds for abdominal obesity differ between different public health agencies [3]. Although waist circumference is known to be affected by sex and race, waist circumference measurements are still considered as a useful anthropometric marker of visceral fat accumulation that contributes to the risk of MetS and obesity [3]. However, measures of the waist circumference may include errors depending on the measurement location or technique. Of note, the prognostic value of waist circumference measurements is poor in severely obese individuals [8]. Therefore, there is a need for an additional anthropometric marker that in predicting disease risk is easy to measure and reliable to address the limitations of current anthropometric indicators for screening disease risks. Neck circumference (NC) is an easy-to-use, low-cost, timesaving, and convenient screening tool. Therefore, it is ideal for routine assessments in primary care clinics and other medical settings [9].

According to previous studies, free fatty acids (FFAs) from the upper subcutaneous adipose tissues had greater association with MetS and CVD risk factors than FFAs found in the abdominal visceral fat [10,11]. NC is an anthropometric factor indicative of upper subcutaneous fat deposits. By measuring fat accumulation around the neck, NC may be a reliable factor in predicting metabolic disease risk [11,12,13]

NC is used to predict the prevalence of cardiovascular disease, obesity, and MetS, including obstructive sleep apnea [14,15,16,17]. Studies over the past decade have shown that NC is independently associated with MetS [16], obstructive sleep apnea [17], and cardiovascular disease [18,19].

Despite the reports of these previous studies, NC was not included in the anthropometric measurement for diagnosing disease risk due to insufficient epidemiologic and population-based studies on the clinical significance of NC. In particular, there were not enough related studies in Korea, as the neck circumference variable was added to the Korea National Health and Nutrition Examination Survey (KNHANES) for the first time in 2019.

As mentioned above, the results of studies evaluating the prevalence of MetS are conflicting due to variability in the diagnostic criteria [20]. Moreover, the prevalence of MetS varies according to ethnicity because genetic factors, age, dietary habits, and socioeconomic factors all contribute to the risk of developing MetS [21]. The prevalence of MetS has been estimated to be approximately 20–25% worldwide [22]. It has been reported that the prevalence of MetS is increasing in a trend similar to those of type 2 diabetes, hypertension, cardiovascular disease, and obesity [22]. Considering the increasing risk of type 2 diabetes and cardiovascular disease worldwide, there is a need for a practical indicator, such as the NC index, that can accurately identify individuals with MetS [2]. The ability to rapidly screen individuals to determine the risk of disease and managing the modifiable factors is important for reducing the global disease burden.

Therefore, in the present study, we analyzed the data from 2019 in the KNHANES to explore (1) the potential relationship between NC and MetS, (2) the optimal NC cut-off point for predicting MetS risk of Koreans and (3) the association between the NC cut-off point and MetS risk.

## 2. Materials and Methods

### 2.1. Study Population

The purpose of KNHANES, carried out annually by the Korea Disease Control and Prevention Agency under the Ministry of Health and Welfare, is to evaluate national representative and reliable statistics on the health level, health behavior, and food and nutrition intake of Koreans, and utilize them as basic data for health policy. To improve the sample representativeness and the estimation accuracy, KNHANES was extracted using the multistage stratified cluster probability extraction method, a complex sample design method.

KNHANES was calculated by reflecting the design weight calculation, nonresponse adjustment, post correction, and extreme weight processing so that the estimate was given to represent the entire population of Koreans. KNHANES was conducted by trained medical staff and interviewers and consisted of a health interview, health examination and nutrition survey. More detailed information on KNHANES is previously described elsewhere [23].

We collected data from 3073 participants aged ≥40 to <65 years among the total 10,859 who participated in 2019 KNHANES. Data of participants were sequentially excluded as follows: 315 participants who had thyroid disease, tumor in the neck, or no response, 33 participants who had extreme (≤500 or >5000 kcal) energy intake, 36 participants who had body mass index (BMI ≥ 39.0 kg/m^2^) outside the normal distribution and WC (≥110 cm) as outliers from the analysis, and 455 participants with no information on NC, MetS diagnosis components, and food intake survey. As a result, the data of 2234 participants were analyzed in our study.

KNHANES was conducted with the approval of the Institutional Board of Korea Centers for Disease Control and Prevention (IRB number: 2018-01-03-C-A), and all participants provided informed consent. This study was approved by the Institutional Review Boards of the Chonnam National University (IRB number: 1040198-210114-HR-003-01).

### 2.2. Neck Circumference and Anthropometric Characteristic Measurements in KNHANES

In KNHANES, to minimize measurement errors, staff underwent mandatory training twice a year including theoretical and practical training. To measure the neck circumference, all participants were seated on a chair with their hips, waist, and back touching the backrest while maintaining a right-angle posture. Next, measurements were taken by trained staff from directly under the Adam’s apple with the participants positioned with their necks upright, their heads parallel to the Frankfort plane, and their arms naturally lowered. From the right side of the participants, the staff member measured up to 1 mm by ensuring that the tape measure (Lufkin W606pm, Lufkin Industries, Inc., Missouri, TX, USA) was perpendicular to the long axis of the neck. As mentioned above, for males, the trained staff measured the NC by palpating the position of Adam’s apple of the subject. For female, the NC was measured by tilting the participant’s head back so that the thyroid cartilage protruded, positioning it, and then the patient staring at the front again. In order to reduce the measurement error, the measurement was repeated twice and the average value was presented.

To measure WC, trained staff measured the midpoint between the bottom of the last rib and the top of the iliac ridge in the midaxillary line on the right side of the participant. The height and weight of the participants were also measured by trained staff according to the KNHANES standard manual. The BMI was calculated as body weight in kilograms divided by height in meters. Grip strength was measured by trained staff starting with the main hand using a Takei digital grip strength dynamometer (T.K.K.5401, Takei, Japan). The measurements were recorded three times with both hands crossed. After each measurement, the participants rested for 60 s.

### 2.3. MetS Assessment and Biochemical Characteristic Measurements in KNHANES

The KNHANES health examination consisted of BP and pulse measurements, blood and urine tests, oral tests, lung function tests, and eye tests.

BP was measured thrice by a nurse after a 5-min rest. In KNHANES, blood tests were performed after checking the fasting time (at least 8 h) by investigating the time of last food intake before blood collection after confirming the participant’s name, age, and identity. The serum levels of total cholesterol, high-density lipoprotein-cholesterol (HDL-C), triglycerides (TG), low-density lipoprotein-cholesterol (LDL-C), aspartate aminotransferase (AST), alanine aminotransferase (ALT), and fasting glucose were measured using a Hitachi automatic analyzer 7600 (Hitachi/Japan). Glycated hemoglobin (HbA1c) was measured by high-performance liquid chromatography using Tosoh G8 (Tosoh G8, Tosoh, Japan). The cholesterol and triglyceride levels were measured by enzymatic methods, and FBG (fasting blood glucose) was measured by Hexokinase UV. AST and ALT were measured by G-glutamyl-carboxy-nitroanilide (IFCC) UV without P5P.

The diagnostic criteria for MetS were based on the NCEP-ATP III MetS diagnostic criteria and the abdominal obesity diagnostic criteria for Koreans published by the Korean society for the study of obesity [24,25]. The criteria for diagnosing MetS were as follows. Participants who met at least three of the following five items were considered MetS-positive.

(1)Abdominal obesity (WC ≥ 85 cm for women and ≥90 cm for men)(2)High BP (diastolic BP ≥ 85 mmHg or systolic BP ≥ 130 mmHg)(3)Hyperglycemia (FBG ≥ 100 mg/dL)(4)Low HDL-C (HDL-C < 50 mg/dL for women and < 40 mg/dL for men)(5)Hypertriglyceridemia (TG ≥ 150 mg/dL)

### 2.4. Assessment of Dietary Intake and Dietary Habits in KNHANES

The food intake survey of KNHANES was conducted with a 24-h recall method by a trained interviewer. In the investigation process, auxiliary tools such as the volume calculation tools, sample models of foods, measuring cups and measuring spoons, were used to enhance the accuracy of the participant’s recall regarding their food intake. In KNHANES, food and nutrient database using food composition tables (Rural Development Administration, 2020) and Dietary Reference Intakes for Koreans (Ministry of Health and Welfare, 2015) were applied to quantify the daily intake of food and nutrients.

In the 8th KNHANES, the same food code was generated for foods with the same raw material. Table 1 shows the food intake converted based on the food code.

To evaluate the dietary habits of participants, we checked variables such as diet frequency, eating out frequency, and the use of nutritional supplements.

### 2.5. Assessment of Other Socioeconomic Characteristics

Participants in KNHANES were interviewed for demographic characteristics such as sex and age, as well as other characteristics such as smoking status, alcohol use, history of diseases, and obstructive sleep apnea.

### 2.6. Statistical Analyses

All analyses were performed using SPSS 18.0 software (IBS SPSS Statistics, Armonk, NY, USA). The data are expressed as mean ± standard error for continuous variables and number (%) for categorical variables. Significant differences between characteristics were verified by Student’s t-tests and chi-squared tests. Pearson’s correlation coefficient was used to evaluate the relationship between NC and MetS risk factors. The optimal NC cut-off point for predicting MetS was determined by receiver operating characteristic (ROC) curve analysis. Subsequently, a multivariable logistic regression model was used to assess the associations between different NC categories (≥cut-off value vs. <cut-off value) and MetS risk and its components (categorical variables). The results were described as odds ratio (OR) and 95% confidence intervals (95% CI).

Logistic regression models for confounding factors that may affect the analysis were adjusted as follows. Model 1, unadjusted. Model 2, adjusted for age group. Model 3, adjusted for age groups, history of diseases, family history of diseases, smoking status, energy (kcal/day), sugar (g/1000 kcal), and sodium (mg/1000 kcal). Model 4, adjusted for model 3 + BMI (kg/m^2^). Model 5, adjusted for model 3 + WC (cm) [9,26,27]. A *p* value < 0.05 was considered statistically significant.

## 3. Results

### 3.1. Characteristics of the Study Participants

The demographic and biochemical characteristics of the participants were collected. The characteristics according to sex are presented in Table 2. There was no difference in the age group distribution by sex. Among those who participated in the 8th KNHANES, data of 2234 participants (men, *n* = 974; women, *n* = 1260) were included in this study. Of the total, 643 participants (28.8%) were diagnosed with MetS, 364 participants (37.4%) for men and 279 participants (22.1%) for women (*p* < 0.001). Variables, such as the history of diseases, obstructive sleep apnea, smoking status, and drinking habit, were significantly different according to sex. The mean grip strength was significantly higher in men (38.8 kg) than in women (22.5 kg) (*p* < 0.001). WC (88.2 cm vs. 80.7 cm; *p* < 0.001) and NC (38.3 cm vs. 32.7 cm; *p* < 0.001) were significantly higher in men than in women (*p* < 0.001). BMI, BP, FBG, HbA1c, TG, AST, and ALT were higher in men than in women (*p* < 0.001), while total cholesterol (TC), LDL-C, and HDL-C were higher in women than in men (*p* < 0.01).

Nutrients and food group intake characteristics of the participants are presented in Table 1. The mean energy intake of participants was 1950.9 kcal/day (men, 2262.3 kcal; women, 1639.5 kcal; *p* < 0.001). Women had a higher intake of carbohydrate (g/1000 kcal), fat (g/1000 kcal), protein (g/1000 kcal), water (g/1000 kcal), sugar (g/1000 kcal), calcium (mg/1000 kcal), phosphorus (mg/1000 kcal), potassium (mg/1000 kcal) and vitamin C (mg/1000 kcal) than men (*p* < 0.05). Sodium (mg/1000 kcal) intake was 1913.1 (mg/1000 kcal) for men and 1792.7 (mg/1000 kcal) for women, and the average of all participants was 1852.9 (mg/1000 kcal) (*p* < 0.01).

In sum, men consumed significantly more food than women in all food groups except potatoes and starches (g/day) and mushrooms (g/day) (*p* < 0.001). In particular, the intake of meat (212.2 g/day vs. 130.3 g/day) and alcoholic beverages (530.6 g/day vs. 243.9 g/day) was significantly higher in men than in women.

### 3.2. Correlation between NC and Risk Factors for MetS

The results from correlation analysis between NC and MetS risk factors by sex are shown in Table 3. NC was positively correlated with BMI (r = 0.809, *p* < 0.001) and WC (r = 0.767, *p* < 0.001) in men. Similarly, in women, NC was positively correlated with BMI (r = 0.770, *p* < 0.001) and WC (r = 0.766, *p* < 0.001). Besides, NC was positively correlated with BP, FBG, HbA1c and TG, and negatively correlated with HDL-C in both (All *p* < 0.001).

### 3.3. Determining the Optimal Cut-Off Point of Neck Circumference for Diagnosis of MetS

ROC curve analysis was used to detect sex-specific NC cut-off values for predicting MetS (Figure 1). NC values of ≥38.25 cm (AUC: 0.759) for men, and ≥33.65 cm (AUC: 0.811) for women were the cut-off values best able to predict MetS. In our study, the NC cut-off value was set as the largest best fit value with the sum of sensitivity + specificity-1 [28].

### 3.4. Association of the Neck Circumference with MetS and its Components

Association between NC (categorical variable) and the risk of MetS and its components was analyzed (Table 4 and Table 5). We found that NC above the cut-off point was associated with WC ≥ 90 cm in men (OR: 1.248, 95% CI: 1.205–1.293; *p* < 0.001) and ≥85 cm in women (OR: 1.276, 95% CI: 1.234–1.319; *p* < 0.001). Furthermore, NC above the cut-off point was associated with high BP in men (OR: 1.041, 95% CI: 1.013–1.070; *p* = 0.004) and hypertriglyceridemia in women (OR: 1.002, 95% CI: 1.000–1.004; *p* = 0.042).

Multivariate linear regression analysis (Table 5 revealed that NC above the cut-off point was significantly associated with an increased MetS risk in both men (Model 5, OR 2.014; 95% CI 1.348–3.008; *p* = 0.010) and women (Model 5, OR 3.650; 95% CI 2.382–5.594; *p* < 0.001) after adjusting for the confounding factors.

## 4. Discussion

In this study, we found that NC was correlated with MetS risk factors (particularly, BMI and WC) in Koreans aged ≥40 to <65 years. We also estimated the optimal sex-specific cut-off value of NC to predict the risk of MetS and found that the estimated value of NC was significantly associated with MetS risk in both men and women, even after correcting for covariates including BMI and WC. In both men and women, those with MetS had a greater NC than those without MetS (data not shown) (39.5 cm vs. 37.5 cm in men and 34.6 cm vs. 32.3 cm in women, *p* < 0.001). A previous study from Thailand, published in 2020 [29], also found that the mean NC was significantly higher in participants with MetS (35.0 cm vs. 31.6 cm in women, *p* < 0.001) than those without MetS. These results support the possibility of using NC to predict MetS.

Recent studies have been conducted to estimate the association between NC and various diseases. For example, a cross-sectional study of 4238 Chinese from the Bao’an District of Shenzhen (southeast China) showed that NC was significantly associated with cardiometabolic disease [18]. In a prospective cohort study of 4000 participants over an average of 10.9 years, Zhang and colleagues concluded that NC may be a preclinical predictive value for coronary heart disease death and congestive heart failure occurrence, and that it may also be an early risk factor [30]. Chen et al. evaluated NC and noted that it was significantly associated with the risk of cognitive impairment, as well as BMI, higher waist to hip ratio, and higher TC in the elderly over 60 years of age [31]. Consistent with these previous studies, our results also showed that NC was significantly associated with MetS and its components. Considering the results of recent studies on the relationship of NC and various diseases, we determined the optimal cut-off point of NC as a predictive risk factor for MetS in Koreans.

Based on the results of this study, the NC value should be considered for the screening of high-risk individuals as part of disease management.

ROC analysis was used to evaluate the accuracy of prediction of MetS using the specificity and sensitivity of the NC. The optimal fit variable was defined as (sensitivity + specificity – 1) as reported in Arnold et al. [28]. That is, the NC value of ≥38.25 cm in men and ≥33.65 cm in women was appropriate to predict the risk of MetS. These results are consistent with findings conducted in Thailand with 201 men and 386 women (NC value of men 38 cm, NC value of women 33 cm) [32]. In a cross-sectional study of 1053 Brazilian adults aged 18–60 years, Stabe et al. [33] found that the optimal NC cut-off value was >40 cm for men (AUC was 0.73) and >36.1 cm for women (AUC was 0.74) (*p* < 0.001) and concluded that NC is a simple tool to identify MetS and insulin resistance. These measurements are relatively higher than those observed in our study [33]. In a case-control study of 215 South Asian Pakistani individuals by Hingorjo et al. [34], the optimal NC cut-off point for determining MetS was ≥38 cm for men (AUC was 0.76) and ≥34 cm for women (AUC was 0.63), which are similar to the cut-off values in our study. AUC of the optimal NC cut-off values for MetS in our study was 0.759 (95% CI = 0.729–0.790) for men and 0.811 (95% CI = 0.782–0.840) for women. AUC is not an absolute value, such that an AUC closer to 1 has better predictive power. As mentioned above, AUCs for NC cut-off values in our study were higher than those reported by Limpawattana et al. [32], Stabe et al. [33], and Hingorjo et al. [34]. This suggests that the disease prediction power of the values from our analysis are more accurate than those of the previous studies.

We applied logistic regression analysis to analyze the association between NC and MetS risk based on the cut-off values estimated by ROC analysis. Compared to a lower NC cut-off value in participants, we observed that a higher NC cut-off value was significantly associated with an increased risk for MetS by 2.014-fold (95% CI = 1.348–3.008, *p* = 0.010) in men and 3.65-fold (95% CI = 2.382–5.594, *p* < 0.001) for women in models fully adjusted for confounding factors and WC (model 5).

In summary, although the NC value for predicting MetS differed among different participants, demographic characteristics, and biochemical factors, the results of our study and previous studies confirmed the association between NC and various diseases and demonstrated that NC may be a useful diagnostic index for early screening of individuals at risk for MetS. Other body measurements such as WC, height, and weight are also easy evaluation tools for identifying metabolic diseases. In particular, WC is a well-accepted tool for screening abdominal obesity and is included in the major diagnostic criteria for MetS [35]. However, there are some limitations to measuring WC that are affected by factors such as posture, meal intake, and breathing. WC is also not able to be measured in pregnant women and individuals with ascites. Although the World Health Organization is updating guidelines to standardize the measurement of WC, there are still limitations in measuring accurately [36]. These limitations have emphasized the need for additional anthropometric indicators besides the commonly used WC.

Reflecting on previous studies and our results, the NC measurement should be considered a valid indicator for predicting disease risk. However, NC measurements cannot be used for patients with thyroid disease risk and neck masses (e.g., benign tumor, malignant and common masses). It is an important predictor of disease risk for participants in whom the WC cannot be measured accurately. Furthermore, NC measurements are easy to perform and do not require the removal of clothing. NC is also a time-saving tool, and thus, appropriate for clinical and research settings [37]. Huang et al. reported that the combination of anthropometric factors (such as WC and BMI) and NC may contribute to a more accurate prediction of disease risk [38] In our results (Table 3 and Table 5), NC was significantly correlated with WC (r = 0.767, *p* < 0.001 in men, r = 0.766, *p* < 0.001 in women) and BMI (r = 0.809, *p* < 0.001 in men, r = 0.770, *p* < 0.001 in women). When comparing the lower and higher estimated NC cut-off points, the higher NC cut-off point was associated with elevated WC (≥90 cm in men, and ≥85 cm in women) by 1.248-fold (95% CI = 1.205–1.293, *p* < 0.001) for men and 1.276-fold (95% CI = 1.234–1.319, *p* < 0.001) for women. Similar to our study, Li et al. demonstrated that NC (r = 0.49, *p* < 0.001), an indicator of upper body fat stored [39], was positively associated with visceral fat (by CT scan) in Chinese adults (177 patients with a mean age of 59 years), especially men. It is widely known that visceral fat is closely related to insulin resistance and cardiovascular disease [39]. These results support that the NC measurement should be recommended as an anthropometric indicator for screening high-risk disease individuals, and that both the NC and WC measurements can be used in the clinical diagnosis of MetS. Our study, as well as a number of previous studies [39,40,41], reported that NC can be an important indicator, and the cut-off point for disease differs according to ethnicity and age. The analysis of NC in Korea is insufficient.

To our knowledge, the current study is the first to identify the optimal NC to predict MetS risk using national data representative of Koreans. However, there are some limitations. First, since this is a cross-sectional study, the causal relationship between NC and MetS cannot be identified. Second, NC is a stable indicator of upper body subcutaneous fat, but since our study analyzed using secondary data, the amount of fat accumulated by radiographs could not be measured.

Of note, our study also has some strengths. This study presented the optimal sex-specific NC cut-off values (≥38.25 cm in men and ≥33.65 cm in women) for Korean adults. It is a representative emerging study that analyzed MetS risk according to the estimated NC cut-off values. Using logistic regression analysis, possible factors that may affect MetS were corrected in detail (models 2–5).

Our study found that NC is a reasonable anthropometric indicator for predicting MetS, a risk condition for cardiovascular disease in Korean adults, and that it can be used as a fast accurate screening tool in the clinical setting. However, considering the limitations of our study, large-scale prospective studies should be conducted in the future. Furthermore, studies investigating how NC contributes to an increased risk of MetS should be performed.

## 5. Conclusions

We used the recent KNHANES data to identify the optimal NC cut-off points for the prediction of MetS risk in Koreans.

Of the 2234 participants, 643 (28.8%) had MetS. NC and, in particular, risk factors for MetS, BMI, and WC, were significantly correlated in both sexes (*p* < 0.001). The optimal sex-specific NC cut-off points were estimated using ROC analysis and were 38.25 cm in men (AUC: 0.759, 95% CI: 0.729–0.790) and 33.65 cm in women (AUC: 0.811, 95% CI: 0.782–0.840). Furthermore, logistic regression analysis was performed to adjust for confounding factors to determine the association between NC and MetS. The results showed that in models 4 and 5, with all confounding factors fully adjusted, as well as BMI or WC, NC was significantly associated with MetS risk in both men and women. Based on our results and the results of previous studies, it was found that NC could be a practical anthropometric measurement for predicting the risk of MetS. Considering that this is a cross-sectional study, a large-scale follow-up study should be performed in future to analyze the risk of MetS by combining the original anthropometric indicators with NC.

## Figures and Tables

**Figure 1 nutrients-13-03029-f001:**
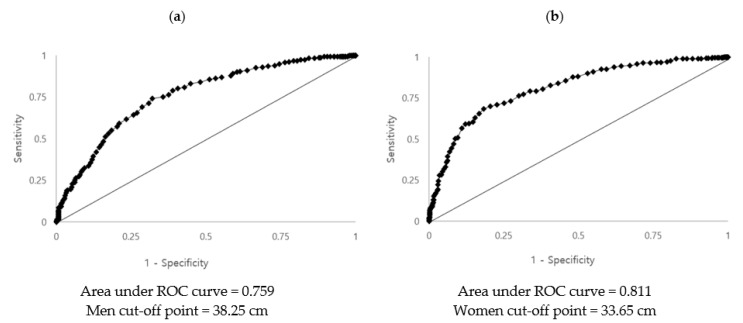
Receiver operating characteristic curve of neck circumference for predicting MetS in men (**a**) and women (**b**).

**Table 1 nutrients-13-03029-t001:** Nutrients and food intake characteristics of participants according to sex.

Variables	Total (*n* = 2234)	Men (*n* = 974)	Women (*n* = 1260)	*p*
Energy (kcal/day)	1950.9 ± 19.5	2262.3 ± 30.2	1639.5 ± 5	<0.001
Carbohydrate (g/1000 kcal)	152.3 ± 0.8	147.5 ± 1.3	157.0 ± 0.9	<0.001
Fat (g/1000 kcal)	22.5 ± 0.3	21.7 ± 0.3	23.3 ± 0.3	<0.001
Protein (g/1000 kcal)	36.8 ± 0.3	36.3 ± 0.4	37.2 ± 0.4	0.050
Water (g/1000 kcal)	566.6 ± 7.4	518.2 ± 9.7	615.0 ± 9.6	<0.001
Sugar (g/1000 kcal)	32.4 ± 0.4	28.5 ± 0.6	36.4 ± 0.7	<0.001
Calcium (mg/1000 kcal)	283.2 ± 3.5	261.5 ± 4.2	304.9 ± 5.5	<0.001
Phosphorus (mg/1000 kcal)	562.3 ± 3.7	540.3 ± 5.1	584.3 ± 4.6	<0.001
Sodium (mg/1000 kcal)	1852.9 ± 21.6	1913.1 ± 28.7	1792.7 ± 27.4	0.001
Potassium (mg/1000 kcal)	1554.9 ± 11.8	1448.5 ± 16.0	1661.3 ± 15.1	<0.001
Vitamin C (mg/1000 kcal)	38.3 ± 1.0	31.1 ± 1.2	45.5 ± 1.4	<0.001
Cereals (g/day)	252.2 ± 2.2	287.3 ± 3.2	217.1 ± 2.4	0.199
Potatoes and starches (g/day)	71.2 ± 2.6	65.9 ± 3.5	76.5 ± 3.5	<0.001
Sugars (g/day)	12.3 ± 0.4	12.8 ± 0.5	11.9 ± 0.5	<0.001
Legumes (g/day)	54.0 ± 1.7	56.4 ± 2.4	51.6 ± 2.3	<0.001
Vegetables (g/day)	292.8 ± 4.2	329.8 ± 6.3	255.8 ± 4.2	0.567
Mushrooms (g/day)	17.7 ± 0.9	17.6 ± 1.2	17.8 ± 1.1	<0.001
Fruits (g/day)	225.3 ± 6.4	217.4 ± 6.4	233.3 ± 10.1	0.640
Vegetable oils (g/day)	6.9 ± 0.2	8.0 ± 0.2	5.8 ± 0.2	<0.001
Meat (g/day)	171.3 ± 3.5	212.2 ± 5.6	130.3 ± 3.5	<0.001
Eggs (g/day)	54.1 ± 1.1	58.0 ± 1.5	50.2 ± 1.3	<0.001
Seafoods (g/day)	126.6 ± 3.0	141.7 ± 4.1	130.3 ± 3.5	<0.001
Beverages (g/day)	163.3 ± 5.4	181.6 ± 7.4	144.9 ± 5.5	<0.001
Alcoholic beverages (g/day)	387.2 ± 18.9	530.6 ± 29.3	243.9 ± 20.7	<0.001

Variables are shown as mean ± standard error.

**Table 2 nutrients-13-03029-t002:** Demographic and biochemical characteristics of the participants according to sex.

Variables	Total (*n* = 2234)	Men (*n* = 974)	Women (*n* = 1260)	*p*
Age groups, *n* (%)				0.157
40–49 years	872(39.0)	381(39.1)	491(39.0)	
50–59 years	881(39.4)	372(38.2)	509(40.4)	
60–64 years	481(21.5)	221(22.7)	260(20.6)	
History of diseases (yes), *n* (%)				
Hypertension	455(20.4)	238(24.4)	217(17.2)	0.010
Stroke	29(1.3)	19(2.0)	10(0.8)	0.022
Cardiovascular disease	38(1.7)	30(3.1)	8(0.6)	0.001
Diabetes	175(7.8)	99(10.2)	76(6.0)	0.012
Cancer	65(2.9)	23(2.4)	42(3.3)	0.012
Depression	109(4.9)	27(2.8)	82(6.5)	<0.001
Obstructive sleep apnea (yes), *n* (%)	15(0.7)	13(1.3)	2(0.2)	<0.001
Smoking status, *n* (%)				<0.001
Current	411(18.5)	353(36.7)	58(4.6)	
Former	536(24.2)	442(45.9)	94(7.5)	
Never	1269(57.3)	168(17.4)	1101(87.9)	
Drinking use (yes), *n* (%)	2045(92.3)	932(96.8)	1113(88.8)	<0.001
Grip strength (right hand, kg)	30.6 ± 0.2	38.8 ± 0.3	22.5 ± 0.2	<0.001
WC (cm)	84.4 ± 0.2	88.2 ± 0.3	80.7 ± 0.3	<0.001
NC (cm)	35.5 ± 0.1	38.3 ± 0.1	32.7 ± 0.1	<0.001
BMI (kg/m^2^)	24.1 ± 0.1	24.7 ± 0.1	23.5 ± 0.1	<0.001
SBP (mmHg)	118.3 ± 0.4	120.3 ± 0.5	116.3 ± 0.6	<0.001
DBP (mmHg)	78.2 ± 0.3	80.4 ± 0.4	76.0 ± 0.3	<0.001
Fasting blood glucose (mg/dL)	102.5 ± 0.7	106.31.0	98.7 ± 0.7	<0.001
HbA1c (%)	5.8 ± 0.0	5.9 ± 0.0	5.8 ± 0.0	<0.001
Total-C (mg/dL)	200.6 ± 0.9	198.1 ± 1.2	203.2 ± 1.3	0.002
HDL-C (mg/dL)	53.0 ± 0.4	48.6 ± 0.4	57.4 ± 0.5	<0.001
TG (mg/dL)	145.6 ± 3.1	177.4 ± 5.2	113.8 ± 2.5	<0.001
LDL-C (mg/dL)	120.2 ± 2.2	113.2 ± 2.6	127.3 ± 3.4	<0.001
AST (IU/L)	25.2 ± 0.3	27.6 ± 0.6	22.7 ± 0.3	<0.001
ALT (IU/L)	24.9 ± 0.6	30.0 ± 1.1	19.8 ± 0.5	<0.001
MetS				<0.001
No	1591(71.2)	610(62.6)	981(77.9)	
Yes	643(28.8)	364(37.4)	279(22.1)	

Continuous variables are shown as mean ± standard error and categorical variables are presented in number (%). *p*-values are derived from the Student’s *t*-test for continuous variables or the chi-squared test for the categorical variables. WC, waist circumference; BMI, body mass index; SBP, systolic blood pressure; DBP, diastolic blood pressure; HbA1c, glycated hemoglobin A1c; Total-C, total cholesterol; HDL-C, high-density lipoprotein-cholesterol; TG, triglycerides; LDL-C, low-density lipoprotein-cholesterol; AST, aspartate aminotransferase; ALT, alanine aminotransferase; MetS, metabolic syndrome.

**Table 3 nutrients-13-03029-t003:** Correlation analysis between NC and MetS components and its risk factors by sex.

	Neck Circumference					
	Men (*n* = 974)		Women (*n* = 1260)		Total (*n* = 2234)	
	r	*p*	r	*p*	r	*p*
Age	−0.079	0.014	0.132	<0.001	0.031	0.147
BMI	0.809	<0.001	0.770	<0.001	0.597	<0.001
WC	0.767	<0.001	0.766	<0.001	0.731	<0.001
SBP	0.131	<0.001	0.165	<0.001	0.197	<0.001
DBP	0.186	<0.001	0.134	<0.001	0.270	<0.001
FBG	0.187	<0.001	0.340	<0.001	0.275	<0.001
HbA1c	0.188	<0.001	0.353	<0.001	0.231	<0.001
Total-C	0.053	0.097	−0.001	0.981	−0.045	0.033
HDL-C	−0.280	<0.001	−0.296	<0.001	−0.401	<0.001
TG	0.199	<0.001	0.317	<0.001	0.343	<0.001
LDL-C	0.041	0.523	0.055	0.534	−0.096	0.062

BMI, body mass index; WC, waist circumference; SBP, systolic blood pressure; DBP, diastolic blood pressure; FBG, fasting blood glucose; HbA1c, glycated hemoglobin A1c; Total-C, total cholesterol; HDL-C, high-density lipoprotein-cholesterol; TG, triglycerides; LDL-C, low-density lipoprotein-cholesterol.

**Table 4 nutrients-13-03029-t004:** Association between NC (categorical variable) and the risk of MetS components by sex.

Study Variables	Total	Men *	Women *
	Crude OR (95% CI) *p*	Crude OR (95% CI) *p*	Crude OR (95% CI) *p*
Increased WC (Abdominal obesity)	1.182(1.157–1.208) <0.001	1.248(1.205–1.293) <0.001	1.276(1.234–1.319) <0.001
High BP	1.058(1.038–1.079) <0.001	1.041(1.013–1.070) 0.004	1.019(0.987–1.007) 0.249
Hyperglycemia	1.012(1.003–1.021) 0.009	1.002(0.994–1.011) 0.606	1.009(0.998–1.020) 0.099
Low HDL-C	0.969(0.957–0.981) <0.001	0.989(0.970–1.009) 0.269	0.988(0.969–1.006) 0.196
Hypertriglyceridemia	1.002(1.000–1.004) 0.012	1.001(1.000–1.002) 0.157	1.002(1.000–1.004) 0.042

* For women, NC < 33.65 cm was the reference; for men, NC < 38.25 cm was the reference. OR, odds ratio; CI, confidence interval; increased WC, waist circumference ≥90 cm in men, and ≥85 cm in women; high BP, high blood pressure ≥130/85 mmHg; hyperglycemia, fasting glucose level ≥100 mg/dL; low HDL-C, high-density lipoprotein cholesterol <40 mg/dL in men, and <50 mg/dL in women; hypertriglyceridemia, triglycerides ≥150 mg/dL.

**Table 5 nutrients-13-03029-t005:** Association between neck circumference (categorical data) and the risk of MetS by sex.

	Total			Men *			Women *		
	OR	95% CI	*p*	OR	95% CI	*p*	OR	95% CI	*p*
Model 1	6.468	4.993–8.380	<0.001	6.902	4.410–8.416	<0.001	12.143	8.533–17.280	<0.001
Model 2	6.513	5.041–8.415	<0.001	6.223	4.477–8.651	<0.001	7.783	5.775–10.490	<0.001
Model 3	5.830	4.702–7.922	<0.001	5.830	4.153–8.183	<0.001	11.538	7.971–16.701	<0.001
Model 4	2.853	2.089–3.896	<0.001	1.899	1.239–2.910	0.003	4.515	2.982–6.836	<0.001
Model 5	1.807	1.272–2.569	<0.001	2.014	1.348–3.008	0.010	3.650	2.382–5.594	<0.001

* For women, NC < 33.65 cm was the reference; for men, NC < 38.25 cm was the reference. OR, odds ration; CI, confidence interval. Model 1, unadjusted; Model 2, adjusted for age group; Model 3, adjusted for age group, history of diseases, family history of diseases, smoking status, energy (kcal/day), sugar (g/1000 kcal), sodium (mg/1000 kcal); Model 4, adjusted for model 3 + BMI (kg/m^2^); Model 5, adjusted for model 3 +WC (cm).

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
