# Peer review of "Neck Circumference as a Predictor of Metabolic Syndrome in Koreans: A Cross-Sectional Study"

_nutrients, 2021, doi:10.3390/nu13093029_

Round 1
Reviewer 1 Report
The manuscript is well-written and adds to the literature on NC measurement and disease risk by identifying population-specific cut-offs based on a well-known population-based cohort. There are some minor corrections required and some references that need to be added to strengthen the manuscript.
Line 38: should be cut-off rather than cu-toff
Line 42: Compared to Europeans, Asians have a much lower value of abdominal obesity, considering the risk of type 2 diabetes. The phrase “lower value” is unclear here – is this a lower quantity or percentage of abdominal adiposity? Please add a reference to support this sentence also.
Line 43-44 : please add a reference
Line 55-56: Although anthropometric method is simple, non-invasive, and reliable measurements to detect disease risk in the primary care environment, there is still no agreement on the standard measurement method. This sentence is unclear. Please check the English. Also, the authors need to clarify what disease or risk they are referring to.
Line 60: excellent in predicting disease risk – remove “excellent” here, since no marker alone would fall under this criteria.
Line 65: According to previous studies, the free fatty acids released from the upper body subcutaneous adipose tissues are higher than those released from abdominal visceral fat and may play an important role in the pathogenesis of cardiovascular disease.
Please include references in this sentence above.
What is meant by higher? Do you mean a greater concentration? Please clarify.
Line 67-68: Please include a reference for this statement : NC is an anthropometric factor indicative of upper subcutaneous fat deposits. This sentence is repeated on line 68 and 69 (one occurrence needs to be removed).
How did you define extreme body mass index (BMI) and waist circumference (WC) for the exclusion criteria? What were the values used? Please include these.
Authors state that measurements were taken directly under Adam’s apple but you recruited both male and female subjects. Therefore, how were measurements standardized for females who do not have an Adam’s apple? Also, for men, the size and position of the Adam’s apple might differ between subjects. Does this affect the measurement? If so, how did you control for this?
Lines 140 and 142 – use abbreviations if they have been referred to previously.
Lines 193-194: why were energy, sugar, and sodium chosen for this model, but fat was not included? Given the contribution of dietary fat to energy intake and its impact on many of the metabolic syndrome parameters, it would make sense to include it. If fat is included, what is the outcome of the model? Does it change the results?
Table 1: what are the question marks for “Never smoked”? please clarify under the table.
Line 235: NC can be used instead of neck circumference
Why was model 5 not adjusted for model 4 + WC? Does this change the results if so?
In the model, was MetS categorical?
Line 309: please include reference 28 for Stabe et al in this sentence:
Compared to participants with a lower NC cut-off value, we observed that participants with a higher NC cut- off value were significantly associated with an increased MetS risk by 2.014-fold (95% CI= 1.348-3.008, p = 0.010) for men and 3.650-fold
The sentence reads as participants were associated with increased risk of MetS, which does not sound correct in English. Consider revising.
Line 329: perhaps the word “environmental” could be replaced with demographic or biochemical factors, since environmental factors were not really assessed in this study.
Line 337: “There is no guideline for a standard WC measurement method” –
There are a number of guidelines for standardizing WC measurements including those from the WHO, NIH, etc. Please see the WHO expert consultation for further information: http://apps.who.int/iris/bitstream/handle/10665/44583/9789241501491_eng.pdf;jsessionid=2864C9B5B4FE9E6AD667F78313064E1D?sequence=1
Please refer to the updated versions of these guidelines rather than stating that there are no guidelines. The limitations of measuring WC accurately can still be referred to and would be one of the main ways NC offers an advantage over WC.
Line 341 and 347: please include “risk” after the word disease in each case.
Lines 353 and 363: please include a reference for this statement: “NC is an indicator of upper body fat stored”
In the discussion, the authors should attempt to explain the possible reasons behind the differences in NC cut-offs between the different studies. Is this related to ethnic / age / other differences between the studies? Is there a need for individual cut-offs for each ethnicity? Please discuss this further.
Reviewer 2 Report
This study analyzed the association between neck circumference and metabolic syndrome (MetS) risk factors in both Korean men and women. The results indicated that NC was positively correlated with many MetS risk factors such as BMI, WC, BP, TG. The novelty of this study is to use ROC curve analysis to determine the optimal cut-off point of NC for the diagnosis of MetS. In addition, the dietary intake and habits were also compared by sex in MetS patients. However, the following limitations need to be addressed.
- Is this study enroll healthy subjects as a baseline control to show the association of NC with MetS risk components?
- It is interesting to study the impact of dietary habits on the increased prevalence of MetS. Again, it needs healthy subjects as a control group.
- Is there a relationship between NC and MetS-associated type 2 diabetes (T2DM) and cardiovascular disease (CVD)? It would help to diagnosis T2DM and CVD.
Reviewer 3 Report
This is a large cohort study concerning accuracy of prediction of MetS using neck circumference instead of waist circumference. The authors adopted a simple but firm statistic methodology, and the results seem to be reliable. I'm not worried about appropriateness of results of the present study including data interpretation. However, since there are already many similar studies, I'm not sure of a scientific merit of this study. The authors should have emphasized the identical and/or different points from the previous studies.
Round 2
Reviewer 2 Report
Accepted as the current format.
Author Response
We would like to thank you for the careful review of our manuscript.
Reviewer 3 Report
I appreciate authors' effort to have improved the manuscript. However, I couldn't find something new even in the revised version. This result might be attributed to my scientific directivity.
Author Response

(The authors gave the same response as above.)
